# Scientific dissemination via comic strip: A case study with SACREBLEU

**Matt Post**
Human Language Technology Center of Excellence
Johns Hopkins University
Baltimore, MD 21211, USA
post@cs.jhu.edu

## Abstract

Comic strips are a naturally appealing medium which provide a visually-attractive means for situating scientific results within a narrative. Although they may not be relevant to all situations and can be time-consuming to produce, they also provide unique opportunities for humor and levity that may be an important tool in disseminating and convincing an audience of the merits of a paper. Furthermore, their decomposition into panels makes it easy to annotate them using standard accessibility tools for images. This paper presents the case for presenting scientific posters as comic strips, using the author's 2018 SacreBLEU poster as a motivating example.

## 1 Introduction

Posters are a common presentation format at conferences. As a medium, a poster is quite different from the paper that often gives rise to it. It is worth enumerating some of the reasons. First, they are experienced differently. A paper is there to be read in detail, at the reader's pace, often multiple times; a poster is typically viewed once, and under the eye of the author. Second, they have different means of appealing to people. A paper's mechanisms for attracting attention include catchy titles, the prominence of the authors, and its citation count; a poster, in contrast, is typically situated in a physical space, and its means of appeal are largely visual. Third, they enable different kinds of conversations. Papers are one-way, and follow fairly formal and rigid presentation structures; in contrast, a poster is fodder for an informal conversation, in which the presenter chooses which pieces to highlight, and can fill in interesting backstory. Finally, and perhaps most fundamentally, they have different purposes. The paper's purpose is to motivate a problem and present a solution, and its authors' goal might be said to be to have an effect on a community's conversation; in contrast, the purpose of a poster is the shorter-term goal of getting people to read its corresponding paper.

Walking through many conference poster sessions, however, it can seem that few people seem to note these differences and to take advantage of the poster's unique purpose and format. Many posters are bland and highly text-driven, with the worst of them literally pasting the pages of the paper into a poster frame. While very high profile papers and prestigious groups may be able to get away with this,[1] many papers and ideas are underperforming on a crucial component of the scientific process of *disseminating* information. This is unfortunate, and the situation has led to appeals such as the #betterposter movement.[2] The best posters manage to attract attention equal to or beyond the weight of their underlying paper. They accomplish this by taking advantage of the unique setting of the medium: temporal, spatial, and visual.

One way to accomplish these goals is to use the graphical medium of a comic strip to tell a paper's story. Such an approach allows the presenter to share the underlying story of a paper, which has the potential to engage an audience at a deep and meaningful level. A comic strip also brings along an advantage in terms of accessibility, because it is amenable to panel-by-panel "alt text".

---

[1] Though anecdotally, I think successful groups are also good at presentation.
[2] See Mike Morrison on Youtube: youtube.com/watch?v=1RwJbhkCA58.

This paper discusses the benefits and challenges of such an approach. As a focal point for discussion, I use a poster I presented at the Third Conference on Machine Translation (WMT) in 2018, colocated with the Conference on Empirical Methods in Natural Language Processing (EMNLP) in Brussels, Belgium. The poster, depicted in Figure 1, is also available online as a vector graphic, together with a panel-by-panel accessible variant.[3]

## 2   ADVANTAGES

There are a number of advantages to presenting a scientific poster in comic form:

- Comics are visually engaging. They stand out in a crowd, and help attract attention amidst a sea of other posters which, at some level, are competition.

- Comics induce a narrative or story element. A narrative also helps situate a work and its implications within the real world. This can help humanize and motivate an otherwise abstract, saturated, or even boring topic.

- The act of condensing a paper into a series of discrete panels is an aid to accessibility. The story is defined ahead of time, and each panel can be annotated with a machine-readable text description.

- Comics can help educate. Papers are often written in a results-first fashion that emphasizes a problem and a proposed solution. In many cases, the connection between the two is lost, because the chronology of the work in the paper is obscured, leaving the reader to wonder how the authors arrived at a particular solution. A comic may choose a story arc that helps establish this connection.

- Comics are an informal medium that provide a layer of levity, making all topics more interesting, and also helping more serious points to go down easier.

## 3   CHALLENGES

There are also many challenges to presenting a poster in this fashion. The first is that drawing in comic form draws on a different set of skills, and even the best researcher may feel inadequate to the task. Construction of a comic is time-consuming, likely moreso than for traditional posters.[4] It's possible that many papers are not amenable to comic form, and attempting to do so could prove tiresome. Finally, widespread adoption of a comic format could limit the portion of its appeal that is due to novelty.

## 4   THE SACREBLEU POSTER

The SacreBLEU poster was extremely well-suited to exposition in comic form. My paper, *A call for clarity in reporting BLEU scores* (Post, 2018), pointed out the problem that BLEU (Papineni et al., 2002), the main metric of the machine translation (MT) community, was underspecified, and that varying unreported parameters could result in wild variance in scores that was larger than gains accepted as indicative of successful new methods. The paper was written in response to frustration that many MT researchers felt in trying to compare results across papers. This naturally suggested a narrative in a researcher goes through this process. I created the comic using the Procreate app[5] on a 2018 iPad Pro. A time lapse of its creation is available on Youtube.[6].

In the story arc, a researcher comes up with an idea, finds it works in her local setting, struggles to compare her local numbers to those in recent papers, becomes frustrated, and is helped by a reified form of the tool. The comic provided many opportunities for humor. The first was the name of the tool, which is a play on the BLEU metric name, an antiquated French curse, and the important, central role the metric plays in MT research. This name suggested the appearance of a genie-like

---

[3]waypost.net/text/sacrebleu-poster
[4]I estimate I spent 20 hours on the SacreBLEU poster.
[5]procreate.art
[6]youtube.com/watch?v=kU4PDTLX-0U

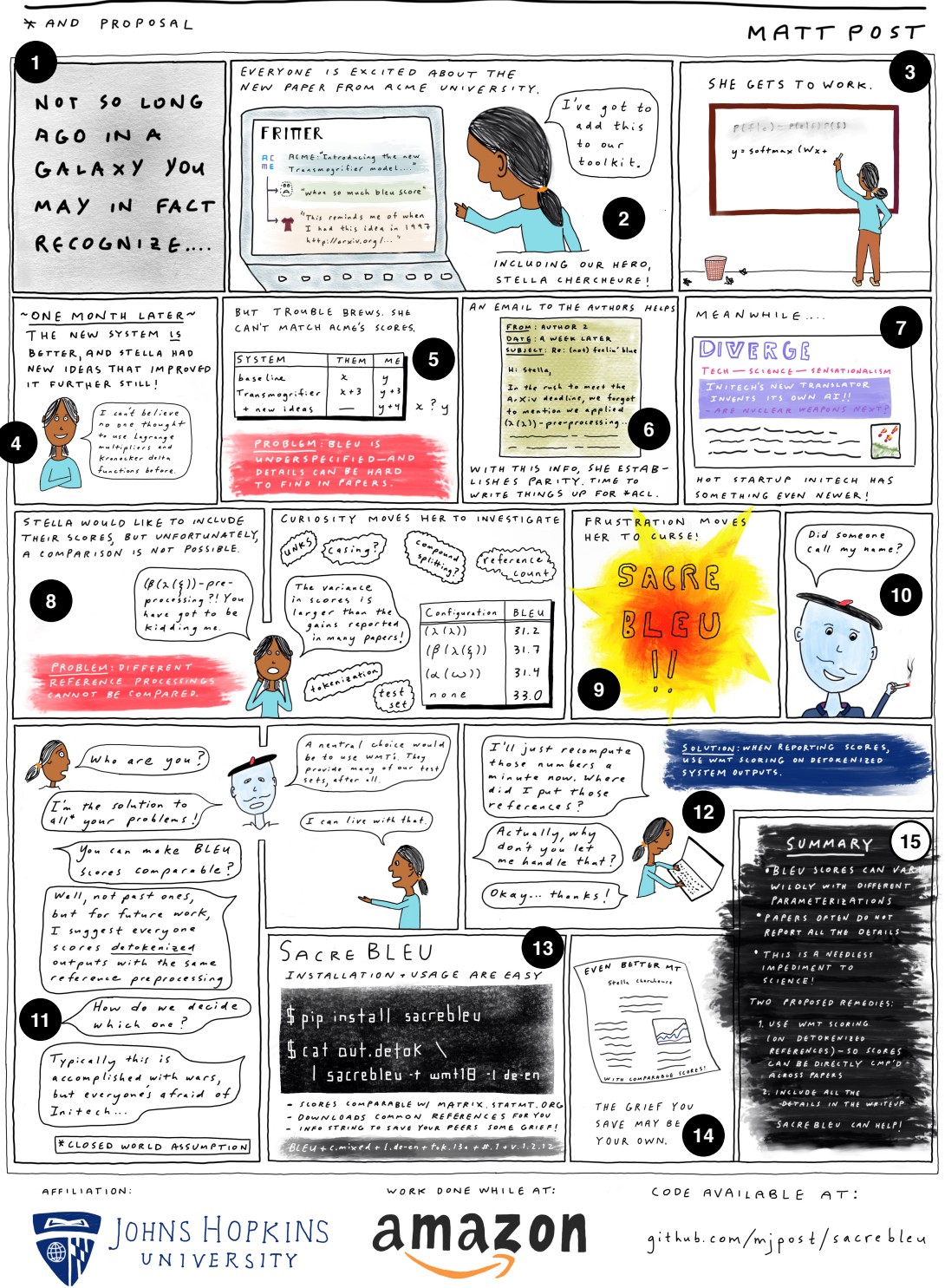

Figure 1: The SacreBLEU poster.

entity invoking American stereotypes of the French. It also employed many other inside jokes, which I enumerate here, because I can't pass up the opportunity:

1. (Panel 2) Acme is the obvious generic commercial super-entity, here invoking Google. The "Transmogrifier" is a reference to a Calvin & Hobbes prop (an aspirational allusion), as well as the Google's Transformer architecture Vaswani et al. (2017). On "Fritter"—the social media network where you waste all of your time—a commenter gushes reverently and an old-timer brings up his own work.[7]

2. (Panel 3) SacreBLEU was introduced in what now feel like the early days of the transition from statistical to neural MT, so Stella is depicted erasing Bayes' rule in favor of a matrix equation.

3. (Panel 4) Stella's note here is a reference to a famous workbook (Knight, 1999) that popularized and made accessible some dense early work in machine translation Brown et al. (1993).

4. (Panel 6). Funny because there is no arXiv deadline.

5. (Panel 7). A nod to the fast pace of research and to tech sensationalism.

6. (Panels 9&10). Stella explodes in frustration and inadvertently summons the SacreBLEU genie.

Using a comic for the poster thus presented a unique opportunity to synthesize many cultural ideas and jokes.

I will note also the serious note that underlies the poster. The problem that SacreBLEU was written to address was, and remains, a serious issue. I've used the term "frustration" in this paper, which might belie the serious annoyance that many researchers felt at what were considered to be sloppy reporting methods by newcomers to the MT community, many of whom were deep learning practitioners applying their techniques to different problems. In my opinion, presenting the poster in comic form took some of the edge off the criticism, adding some fun and deprecation to an important lecture. It's possible that this was helpful in convincing people to swallow the bitter core of the paper, helping establish the tool within the research community. Of course, this is impossible to gauge. For one, I was not alone in making this point, and nor was I the first, and SacreBLEU's other features also likely helped with adoption.[8]

## 5 SUMMARY

It is common for paper authors to ignore the unique opportunities available when presenting information via poster. This paper has presented some reasons to consider presenting posters in a particular graphical form, that of a comic strip, using as an example a poster I presented in 2018. There are challenges to using comics on posters, and they may not be relevant in every setting. But comics are an important visual medium, providing a way to engage an audience with a fresh perspective, in a way that can add humor and levity on top of the cold progress of science and engineering. In addition, both the process and the end result of the comic are an aid to accessibility; the process helps condense the story into discrete steps, and the medium is amenable to computer-readable image annotations. I found the medium to be fun to work with, and public and private feedback suggested that the poster was appreciated and well-received. Although it is impossible to quantify, I suspect the success of the poster may have contributed to the adoption of the procedures and toolkit that were promoted in the paper.

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
