# OpenReview forum: "Scientific dissemination via comic strip: A case study with SacreBLEU"
_ICLR.cc/2021/Workshop/Rethinking_ML_Papers/Exhibit_and_Workflow — Rethinking ML Papers - ICLR 2021 workshop Poster_

### Official Review · Reviewer_nqae · 2021-03-28
**[Strong accept] The proposed use of comic strips in conference posters is novel, and its merits have been clearly demonstrated via the case study - The poster is engaging, humorous and informative.**

**Accessibility:**

Score of 5 (Exceptional): Submission identifies and articulates accessibility matters, provides justifications for the proposed paradigm, and declares the limitations.

**Litreview:**

Score of 4 (Strong): The submission directly differentiates itself from previous works and formats.

**Problemstatement:**

Score of 4 (Strong): The submission sets a very strong example of how to address the problem, which should be relevant to the workshop themes.

**Relevance:**

Score of 5 (Exceptional): Like (4) but does so with multiple themes of the workshop.

**Results:**

Score of 5 (Exceptional): Submission has an excellent design and all criteria are addressed. Conclusions, practical/theoretical implications are well articulated.

**Reviewerconfidence:**

5
As the creator of scientific comic books myself, I deeply appreciate the proposed framework to use comic strips in poster presentations. Comics - whether shorter strips or long form books - are a novel pedagogical instrument and greatly aid in the dissemination of scientific scholarship. To the best of my knowledge, their proposed use in conference posters is novel, and its merits have been clearly demonstrated in the proposal - The poster is engaging, humorous and informative.

**Reviewtext:**

The proposed framework employs comic strips as an accessible medium of scientific dissemination - both in the form of an engaging narrative instrument and one that boosts digital accessibility due to the modular nature of a comic strip. The case study using SacreBLEU aptly demonstrates the strengths of the framework - it captures nuances of the scientific method that are usually omitted in papers, including inspiration, negative results, roadblocks and the overall journey of the scientist. These nuances are (arguably) important to situate the scientific contribution and hence a medium - such as the proposed comic strip - that includes them in the presentation of results makes the overall contribution stronger. The arguments around the use of levity and humor to educate the reader and to incentivize people to engage with a piece of scientific enquiry are also well demonstrated through the case study.

**Score:**

Strong accept: The reviewer has a strong enthusiasm to apply the proposed framework in their work.

---

### Official Review · Reviewer_Lsyn · 2021-03-31
**Interesting idea for poster presentations via comics**

**Accessibility:**

Score of 3 (Neutral): Submission proposes methods to improve accessibility, but the level of intended accessibility is not well-articulated. Also, the limitations and exceptions are not stated.

**Litreview:**

Score of 2 (Needs Improvement): The submission leaves out prominent examples of previous work in the area.

**Problemstatement:**

Score of 4 (Strong): The submission sets a very strong example of how to address the problem, which should be relevant to the workshop themes.

**Relevance:**

Score of 4 (Strong): The submission directly addresses a theme of the workshop, and does so in a very professional manner.

**Results:**

Score of 2 (Needs Improvement): Submission shows a poor level of clarity, novelty, coherency, and interactivity.

**Reviewerconfidence:**

3

**Reviewtext:**

The authors propose to use comics as theme to present conference paper posters to effectively disseminate scientific information. They argue that comics are visually engaging, induce a narrative element and informal which could potentially make it engaging for the readers.

Strengths

- Author discuss both the advantages and disadvantages extensively.
- They also provide an example of a presentation in the form of a comics which is engaging.

Weakness

- It would have strengthened the paper if the authors had performed a brief human study to quantify the engagingness.
- Might become a hassle to the authors as they already the tasks of creating presentations, video recordings, and posters for presenting in another venue.

Questions

- Given the informal nature of the presentation in comics, have the authors explore the chances of them being offensive inadvertently?

Typos
- Page 4, point 5 - "pasce" --> "pace"


Overall the paper is well written and easy to follow. I feel that comics are a form of presentation could be engaging as long as there are other alternative engaging forms of narrative presentations of scientific contents.

**Score:**

Accept: The reviewer believes the submission provides a novel and reliable scheme to improve science communication but needs improvement.

---

### Meta-Review · Area_Chair_gRa7 · 2021-03-31

**Recommendation:** Accept
**Confidence:** 5

**Metareview:**

I really like the idea of "comic strips as an accessible medium of scientific dissemination - both in the form of an engaging narrative instrument and one that boosts digital accessibility due to the modular nature of a comic strip." I found the exhibit to be engaging as well. Both reviewers are in agreement as well.

I recommend acceptance. I strongly authors to include feedback from reviewers especially in covering relevant related work,  and if feasible to formalize the impact through a quantitative measure.

---

### Decision · Program_Chairs · 2021-04-01

Accept (Poster)